# Indoor Environmental Quality Assessment of Train Cabins and Passenger Waiting Areas: A Case Study of Nigeria

John Omomoluwa Ogundiran [1,*] , Jean-Paul Kapuya Bulaba Nyembwe [1] ,
Anabela Salgueiro Narciso Ribeiro [2] and Manuel Gameiro da Silva [1,*]

[1] Department of Mechanical Engineering, ADAI, University of Coimbra, Rua Luís Reis Santos, Pólo II,
3030-788 Coimbra, Portugal; kapuyanyembwe@efs.uc.pt

[2] Department of Civil Engineering, University of Coimbra, Rua Sílvio Lima, Pólo II,
3030-790 Coimbra, Portugal; anabela@dec.uc.pt

* Correspondence: johnogundiran@efs.uc.pt (J.O.O.); manuel.gameiro@dem.uc.pt (M.G.d.S.);
Tel.: +351-916104856 (J.O.O.); +351-917362415 (M.G.d.S.)

**Abstract:** The adequacy of the indoor environmental quality (IEQ) in mass transit microenvironments is crucial to the well-being of exposed commuters. By 2050, many developing tropical countries will host even more megacities, which will feature an increase in people mobility and higher occupancy density. The paucity of IEQ studies, the technology gap, and inadequate policy measures to assure safer and sustainable mobility in many developing tropics have reinforced the current study objective. Also, the recent COVID-19 pandemic has highlighted the IEQ links and risks to health in transport, which, given the climate peculiarities, transport reforms, and huge commuter traffic in Nigeria, inform the study motivation. The indoor air quality ($CO_2$, PM, VOCs, $NO_2$), thermal, acoustic, and visual environments were objectively assessed in train passenger cabins and waiting areas, during 15 trips in the dry and rainy seasons in Nigeria. The results were analyzed by following the IEQ requirements defined in the ISO, CEN, ASHRAE, and SAE standards. The results indicate gaps in the IAQ (inadequate ventilation in 9 trains), defective thermal comfort (9 trains), exceedance in the PM limit ($PM_{10}$: 47.9–115 μg/m$^3$, $PM_{2.5}$: 22.5–51.3 μg/m$^3$), noise ($L_{eq}$ range: 64–85 dBA), and low illuminance levels (10 trains), hence the need for IEQ, interventions, stakeholder awareness, and broader IEQ studies on transport cabins in these regions.

**Keywords:** indoor environmental quality; thermal comfort; indoor air quality; particulate matter; noise; visual comfort; trains; Nigeria; developing tropical countries

## 1. Introduction

In many cities, a significant part of mobility needs are met via mass transit on trains and buses, hence the need to minimize the imminent and potential risk associated with the poor indoor environment quality (IEQ) in transport vehicles is necessary to ensure comfort, well-being, health, productivity, and safety.

IEQ refers to the condition of an indoor environment concerning the air quality and comfort parameters such as the indoor air quality (IAQ), thermal comfort, visual comfort, acoustic comfort [1], and ergonomics, including vibration and harshness (in the case of transportation means). These parameters that jointly influence the IEQ should be assured by following regulations and standard requirements depending on the relevance and applicability to the indoor environments.

By 2050, most megacities will be hosted by developing tropical countries, suggesting that these regions will be characterized by significant population density and high mobility traffic. Moreover, the existential high level of outdoor pollutants in several African cities [2] informs the need to ensure an adequate IEQ in these regions since outdoor environmental conditions influence the IEQ of indoor space. Therefore, the need to ensure safer, adequate, energy-efficient, and sustainable transportation in developing tropical countries is essential.

Although modern people spend a greater part of their time in buildings (including offices, shopping centers, schools, homes, religious buildings, hospitals, etc.), a significant amount of time is also spent in transport microenvironments such as cars, buses, trains, aircraft, and other vehicles. Furthermore, mass transit vehicles, such as buses and trains, are prone to a high risk of epidemic outbreaks and health safety [3], coupled with the risk of thermal discomfort from high solar radiation penetration of vehicle cabins [4], thus necessitating adequate IEQ measures and interventions in developing tropics.

Only a few studies have evaluated the IEQ parameters of buildings and mobile indoor spaces in the developing tropics [5], which contributes to justifying the reason for the current study. In the following section, a few studies of the IEQ parameters in train transport microenvironments are now referenced according to the specific parameters investigated, unique findings, and recommendations where applicable. Moreover, in the current study, attention to IEQ assessments of intercity trains during real-time travels in Nigeria has been presented, analyzed, and discussed according to the relevant global IEQ requirements.

Literature review:

Several factors have been identified that impact passenger well-being and comfort in trains including indoor climate parameters like the air and radiant temperatures, air velocity, humidity, and air exchange rate while other aspects such as noise, vibration, barometric variation, light and shadow, colors, and odors are important for achieving passenger comfort during travels [2]. Remarkably, a recent study has evaluated cultural differences in conceptual models of ride comfort in high-speed trains, comparing four viewpoints on the effect of intercultural variables, and classifying critical effects as common (objective effects) and uncommon (subjective effects) [3]. This approach to evaluating ride comfort highlights the peculiarities of occupant behavior that can influence the perception of overall comfort in transport microenvironments considering that culture influences behavior. In developing African countries such as Nigeria with diverse cultural spectra, IEQ studies might need to investigate these influences. There is a global concern about climate change, sustainability, and energy efficiency. Improving transport infrastructure and technology, reducing carbon emissions through energy-efficient alternatives, and the indoor environments of vehicles are concerted efforts for human well-being and safety. Also, the impact of the recent COVID-19 pandemic has informed several reforms and awareness for the minimization of the risk of infectious disease transmission in transport cabins. In developing sub-Saharan countries, several factors contribute to poor IEQ, including environmental policy gaps [4,6], paucity of scientific studies [7,8], poor transport infrastructure [9], alarming environmental pollution [4], poor IEQ awareness, and socioeconomic challenges. As these regions have a high occupancy density in transport vehicles [8], assessing the IEQ conditions is essential to ensure minimal health and comfort risks. Furthermore, only a few studies have investigated IEQ parameters in the developing tropics [7]. Also, there was no published study found on IEQ assessment in trains in Nigeria but two studies on thermal comfort and ventilation in buses by Kamiyo [10] and Odekanle et al. [11] assessed the particulate matter (PM) exposure of commuters in different transport modes. However, studies have investigated the IEQ parameters of trains in other regions.

Thermal comfort: Although thermal comfort is a subjective mental state of occupants' expression of the thermal environment [12], it can be quantitatively determined by evaluating the combined effects of six critical parameters in indoor environments including four physical parameters such as the air temperature ($T_a$), mean radiant temperature ($T_r$), relative humidity (RH), air velocity, and two occupant-related parameters such as clothing and human activity level or metabolic rate [13]. The operative temperature ($T_O$) is representative of the occupant's physiological response to the indoor thermal environment and can be used to evaluate thermal comfort, which can be correlated on a thermal sensation scale to assess the degree of occupants' dissatisfaction or satisfaction with the indoor thermal conditions [14]. Predicted mean vote (PMV) and percentage of persons dissatisfied (PPD) indices are used to evaluate the thermal sensation from the thermal parameters measured

on each train trip in the cabin. The PMV–PPD model was proposed by P.O Fanger in 1970 [15] and adopted by ISO 7730 [16]. Furthermore, it applies to conditions in which the metabolic rate has not exceeded 4 Met [17] such as in offices, hospitals, vehicles, and trains because occupants are in sedentary positions and have minimal activities.

In Tehran, Abbaspour et al. objectively assessed the thermal environments of metro stations and carriages to optimize passenger thermal comfort, and their conclusions suggested that thermal comfort was acceptable, even for the warmest period of the year [18]. In Taiwan, the findings of Lin et al.'s assessment of the thermal perceptions and adaptations of 2129 passengers exposed to short (less than 30 min) and long-haul (more than 60 min) journeys in air-conditioned buses and trains have attributed thermal discomfort to high temperatures, strong solar radiation, and low air movements, while passenger thermal adaptive behaviors were comparatively distinguished in the journey types [19]. The impact of solar radiation on thermal comfort and sensation in high-speed trains was recently studied by Yang et al. Their findings from the evaluation of PMV, PPD, and persons dissatisfied (PD) affirm that solar irradiation infiltration into the cabins increases the thermal load and discomfort whereas the use of roller curtains was recommended to improve the thermal environments and enhance uniformity in the thermal environment [20]. Chen et al. investigated the IAQ of three types of train compartments with varying conditions of speed, outdoor parameters, and window setting, with the conclusion being that more fresh air improves the IAQ [21]. Ye et al. subjectively evaluated the thermal comfort and air quality of 91 train travelers in long-distance passenger rail cars in China, revealing that the IAQ was not as satisfactory as thermal comfort since a cumulative of 76% of passengers wanted improvements in the IAQ regarding fresh air and air velocity [22].

Indoor air quality: Several air pollutants can originate internally or infiltrate vehicle microenvironments, including $CO_2$, carbon-monoxide (CO), volatile organic compounds (VOCs), PM, black carbon (BC), and other volatile inorganic compounds ($VIC_S$) like no, which have been studied regarding vehicle IAQ. In Malaysia, the findings by Masyita et al., assessing the IAQ in trains using measurements and subjective evaluations of 129 persons, were that the $CO_2$ and $PM_{10}$ levels exceeded the acceptable limits, leading to recommendations for IAQ interventions and stakeholder awareness including the need for further IAQ studies [23]. Ongwandee et al. investigated commuter exposure to VOCs (benzene, toluene, ethylbenzene, and m,p-xylene) in four public transport modes in Bangkok, and their comparative finding was that the VOC levels in the in-sky train were statistically lower than for AC/non-AC buses and boats, suggesting that the elevated levels of the sky trains might be reasons for the lesser infiltration of the pollutants [24]. In Singapore, Xiao reported poor IAQ ventilation in the train cabin during passenger peak times on the North–South Line and East–West Line, and passenger thermal discomfort on the North–East Line during less cabin occupation [25]. Similarly, Li et al. seasonally assessed the IAQ parameters in passenger cars of the Beijing Ground Railway Transit System, using mixed methods, leading to the conclusions that the IAQ was acceptable but highlighted the variations in the IAQ due to peak times and passenger density [26]. Chan et al. investigated commuter exposure to PM in various transport modes in Hong Kong, including three railway types, and the findings suggest that the transport mode and ventilation system have a significant influence on the PM level.

Furthermore, railway transport and air-conditioned cabin vehicles are recommended as a substitute for non-air-conditioned vehicles, whereas the highest PM (175 mg/m$^3$) levels were recorded in trams, being 3–4 times more than in trains [27]. Nasir et al. investigated PM pollution in transport microenvironments in the UK, leading to results that showed higher mean PM levels in non-air-conditioned train coaches whereas, during passenger peak times, the PM levels were higher in air-conditioned coaches, highlighting the influence of the peak time (high occupancy density) and PM resuspension, resulting in higher levels in the assessed coaches [28]. Meanwhile, Bai et al.'s assessment of the IAQ in three types of air-conditioned train compartments reported a poor IAQ for all but found comparatively better conditions in the lower-speed train (80 km/h) than in the higher-speed train

(120 km/h) with the conclusion that a fresh air supply is crucial to the IAQ [29]. Russi assessed the IAQ parameters in different transport vehicles including trains in Lisbon. Their findings suggest the PM levels did not exceed the standard limits of Portugal, WHO, and ASHRAE but in comparison to the levels recorded in buses and cars, the PM and aerosol exposure concentrations were lowest in trains while the $CO_2$ levels in all transport modes were linked to the occupancy density in the cabins [30].

Other IEQ comfort parameters: Other IEQ parameters that impact overall passenger comfort have been investigated, such as acceleration effects, vibration, seat static comfort [31–33], and aural pressure effects, highlighting the experience of motion sickness and discomfort in passengers. Significantly, Peng recently reviewed and discussed the main comfort parameters in passenger trains, classifying them into six parameters: lighting, noise, static comfort, vibration, thermal, and aural pressure [34,35]. Xu et al. [36], using a multidimensional approach to assess and evaluate the visual comfort of five subway line cabins, compared an objective measurement with subjective perceptions, exploring five aspects of the visual environment including the effect of the seating layout on visual performance, vertical and horizontal illuminance, spatial brightness, correlated color temperature preference, and the glare uniformity rating in the investigated cabins. Their conclusion implied that there was a measure of compliance with the Chinese and EU requirements regarding illuminance levels. Whereas some studies have investigated links between whole-body vibration (WBV) to metabolic rate, suggesting that an increase in WBV can increase the metabolic rate [37,38]. Meanwhile, changes in the metabolic rate can affect the passenger thermal sensation of the thermal environment [39]; therefore, minimizing vibration and acoustic discomfort in transport cabins can enhance the IEQ conditions.

The main findings from the literature reviewed highlight that most studies have evaluated the indoor climate of train passenger compartments using mixed methods of objective measurements and subjective assessments. Passenger discomfort has resulted more from thermal, visual, and vibration comfort issues but more studies have found unacceptable levels of PM in the cabins. Finally, besides PM infiltration and inadequate fresh air supply, the factors commonly discussed in the reviewed literature were particle resuspension, peak time (high passenger density), strong solar radiation, and low air movements.

## 2. Materials and Methods

Study area: Rail transit infrastructure is still developing in Nigeria, although the Nigerian Railway Corporation (NRC) is 112 years old. It operates a network of less than 4000 Km including single- and double-tracked lines of 150 km from Lagos to Ibadan. However, the rail infrastructure development is progressive with plans to ensure a functional connection within cities such as the light rail project for the Lagos rail mass transit system [40], and between cities such as Lagos, Ibadan, Ota, Calabar, Abuja, Kaduna, Warri, Itakpe, Kano, Jigawa, and Katsina, and even across the nation into Niger [41,42].

Figure 1 shows the railway travel path of the investigated trips as well as field survey photos in the train station and passenger car. The study area spans 156 km of railway or trains traveling from Lagos metropolitan city to Abeokuta and Ibadan metropolis. The main system of transport for these travel routes includes trains, mini-buses, medium coach buses, saloons, and wagon vehicles. The train travel was on the Lagos–Abeokuta–Ibadan rail service operated by the Nigeria Railway Corporation (NRC), which commenced in June 2021. The trips occur from morning and late afternoon, with the departing and returning operations from Lagos and Ibadan railway stations. There are four mains departing and stopping stations in both directions including Mobolaji Johnson Station (Lagos), Babatunde Raji Fashola Station (Lagos), Wole Soyinka Station (Abeokuta), Samuel Ladoke Akintola (Ibadan) Station, and Obafemi Awolowo Station (badan). The longest travel time spans an average of two hours, executing an average of four trips per day according to the Lagos–Ibadan Train Service (LITS). Records show that the passenger traffic has exceeded 30,000 persons monthly since its commencement in the year 2021. The trains consist of executive and regular passenger interior coaches; however, the current study has conducted

measurements of the IEQ only in the regular passenger interior areas/cabins during the reported trips, given that passenger occupancy is typically higher in the regular cabins of no less than 80 persons per coach in capacity. The mobility traffic potential of these trains is high given the high population and socioeconomic activity of Lagos, including its surrounding cities such as Ibadan and Abeokuta. Again, these revamped railway systems are part of an effort by the government, in a private–public partnership agenda, to ensure better transportation and its availability; similarly, other intra-city train transport such as the Lagos blue and red line railway lines are being commissioned. Furthermore, the intercity railway network spans the routes of the entire country, primarily to serve commuters as well as freighting. Therefore, ensuring an adequate IEQ in these trains in developing tropics will contribute to achieving healthier, more comfortable, safer, and sustainable cities, in direct or indirect alignment to the sustainable development goals 7, 9, 11, and 13.

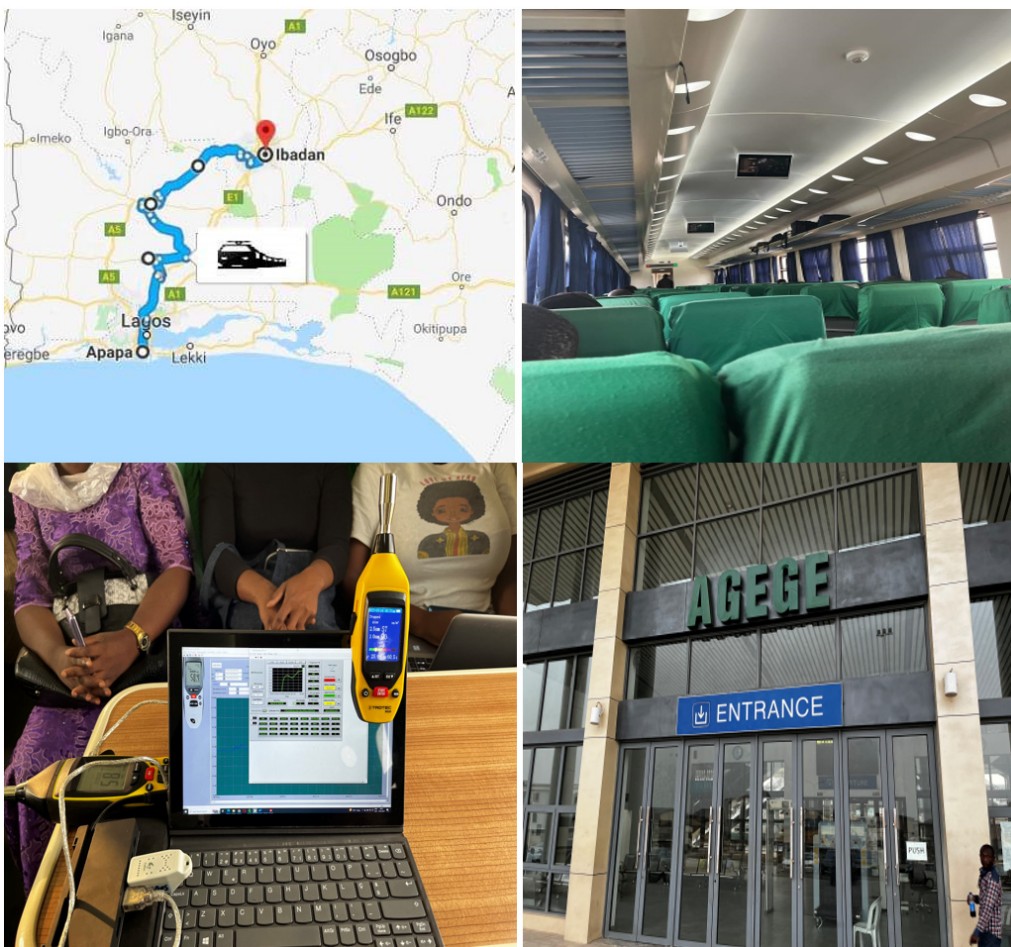

**Figure 1.** Map excerpt showing study area and railway route [43] and photos of the field survey.

Field survey: The field measurements were conducted during real-time travels between February and May 2023 in newly deployed air-conditioned trains for the investigated route. The measurements performed during the trips in February were for the dry season while those in May were considered rainy season assessments. A total of 15 trips were investigated using a calibrated IEQ multiprobe device, connected to a laptop computer via the USB port, which enables easy connection with the computer, ensuring data acquisition and visualization process accordingly [44,45], as shown in Figure 2. Also, a class 2 Trotec SL400 sound meter was used to measure and record the A-weighted sound pressure level. A particle counter, TROTEC BQ20 device, was used to measure the PM2.5 and PM$_{10}$ concentrations during the IEQ assessments in the train cabins. The IEQ multiprobe sensors were

calibrated via the following procedures for these reference types of equipment: a Bruel & Kjaer 1212 Thermal Comfort Meter (operative temperature reference); a Trotec DL200X Data Logger (relative humidity reference); a Trotec DL200L Data Logger ($CO_2$ concentration reference). These values were obtained for the expanded uncertainties (coverage factor = 2, 95% probability) of the measured variables: operative temperature, $T_O = \pm 0.2$ °C; relative humidity, RH = $\pm 1$%; $CO_2$ concentration = $\pm 35$ ppm. For more details regarding the calibration, range, and settings of the measuring equipment used in the objective campaign, see Table S1 in the Supplementary Material. It is noteworthy to indicate that the IEQ multiprobe device used in the current study measures the operative temperature, $T_O$ (°C), by default, considering the air temperature ($T_a$) and radiant temperature ($T_r$) parameters. It was also included to ensure consistent metrological data from the IEQ multiprobe device and complementary calibration software. Table 1 presents information regarding the travel routes, sitting information, and the season during which the measurements were taken.

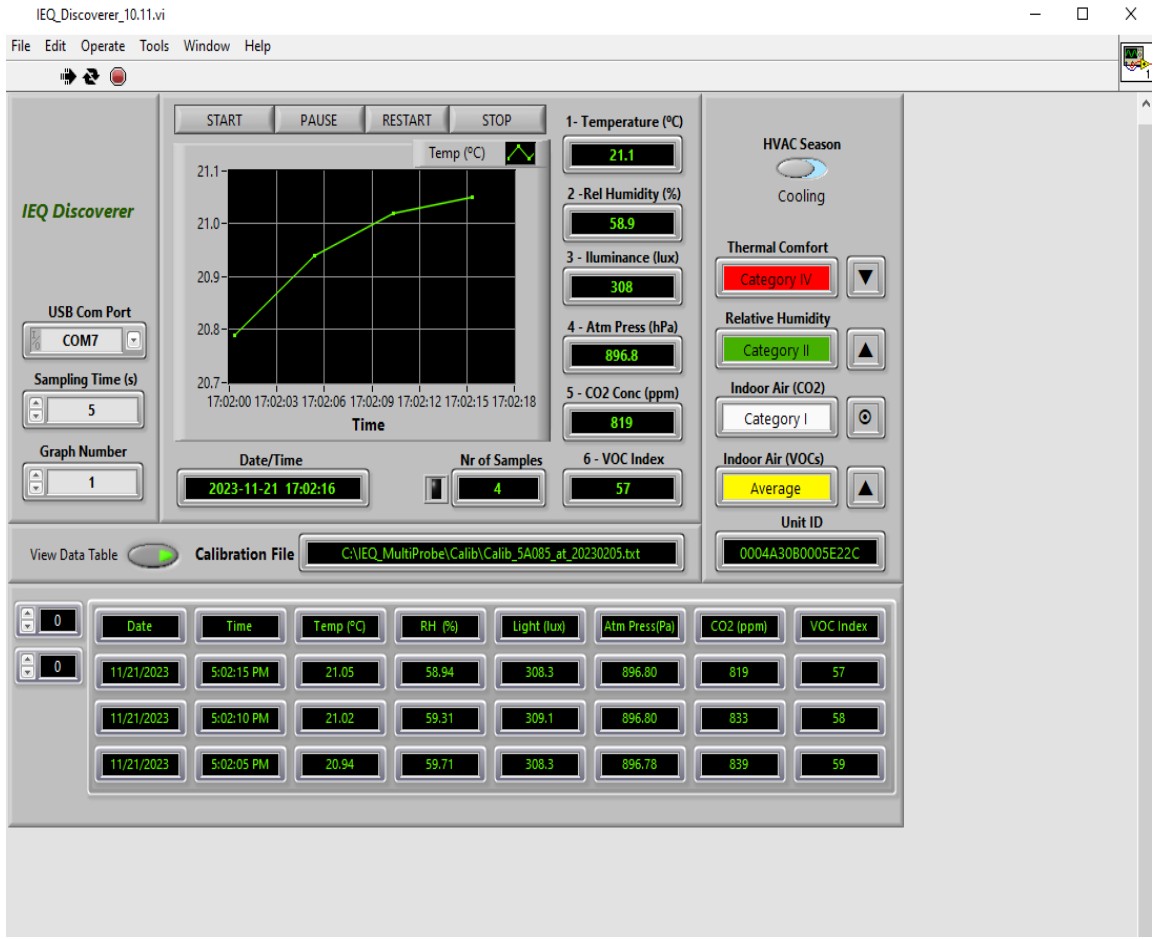

**Figure 2.** Screenshot imagery of the typical data logging and visualization interface of IEQ multiprobe device.

**Table 1.** All train trips, routes, seat and cabin details, travel dates, and seasons.

| Train Trip | Route | Operator | Day-Month | Coach-Seat | Season |
|---|---|---|---|---|---|
| 1 | Ibadan–Lagos | NRC | 16-Feb | C8_40 | Cooling |
| 2 | Lagos–Ibadan | NRC | 21-Feb | C4_10 | Cooling |
| 3 | Ibadan–Lagos | NRC | 21-Feb | C5_85 | Cooling |
| 4 | Lagos–Ibadan | NRC | 23-Feb | C6_41 | Cooling |
| 5 | Ibadan–Lagos | NRC | 23-Feb | C5_34 | Cooling |
| 6 | Lagos–Ibadan | NRC | 05-May | C4_20 | Cooling |

**Table 1.** *Cont.*

| Train Trip | Route | Operator | Day-Month | Coach-Seat | Season |
|---|---|---|---|---|---|
| 7 | Ibadan–Lagos | NRC | 05-May | C6_2 | Cooling |
| 8 | Ibadan–Lagos | NRC | 06-May | C7_18 | Cooling |
| 9 | Lagos–Ibadan | NRC | 06-May | C8_88 | Cooling |
| 10 | Ibadan–Lagos | NRC | 08-May | C5_12 | Cooling |
| 11 | Ibadan–Lagos | NRC | 09-May | C4_5 | Cooling |
| 12 | Ibadan–Lagos | NRC | 12-May | C6_34 | Cooling |
| 13 | Lagos–Ibadan | NRC | 12-May | C6_52 | Cooling |
| 14 | Lagos–Ibadan | NRC | 13-May | C6_41 | Cooling |
| 15 | Ibadan–Lagos | NRC | 13-May | C4_03 | Cooling |

The IEQ multiprobe was used to obtain data for the $CO_2$ levels in ppm, operative temperature ($T_O$ in °C), relative humidity (RH in %), atmospheric pressure (Pa), and lighting levels (lux) including an embedded IAQ index for measuring VOCs. The measurements were taken in a passenger seat at a height of about 110 cm above the cabin floor whereas the IEQ multiprobe sensors functioned omnidirectionally. This allowed for the calculation of the mean values of all the physical parameters according to EN 13129 [46]. In addition, a handheld FLOW air quality measuring device, from Plume Labs, was used to measure and register $PM_{2.5}$, $PM_{10}$, VOCs, and $NO_2$. The air quality index (AQI) data acquired via the Flow device were automatically logged wirelessly via Bluetooth connection to a mobile phone during the trips, with the possibility of access via the mobile app interface and otherwise exported via email in Excel format. The timestamps in the CSV files exported allowed for the easy identification and collection of the relevant data. In the current study, the PM was investigated during five trips carried out during the dry season on account of permits and logistics. Furthermore, HVAC applications in both seasons were identified as applicable to cooling in this region as outdoor temperatures generally exceeded 25 °C. The consideration for the metabolic rate was 1 met [47] and 0.1 m/s for air velocity as passengers were typically in sedentary positions. For clothing, an average clothing insulation (Icl) of 0.6 clo was used for both seasons surveyed. The measured environmental data and the assumed parameters were used to calculate the predicted mean vote (PMV) thermal comfort index. The air quality conditions have been discussed based on the measured parameters of the $CO_2$, VOCs, and PM concentration levels. Noise levels were measured using a TROTEC SL400 (Class 2) sound level meter. The noise equivalent level was calculated for each trip surveyed, whereas these noise levels have been evaluated considering the requirements of relevant indoor noise standards and regulations from Nigeria, the EU, the USA, and Japan, including the Occupational Safety and Health Administration (OSHA). Furthermore, calculations and computations have been made via Microsoft Excel, and the thermal comfort indices using the PMV–PPD spreadsheet calculator of Gameiro da Silva [17], while indoor climate parameters have been categorized and evaluated by following the requirements of the EN 16798-1 standard [48]. National railway standards such as TB/T 1951:1987 [49], GB/T 12817:2004 [50], UIC 553:2004 [51], and BS EN 14750–1:2006 [52], addressing the thermal environments in trains, have mainly prescribed temperature ranges, humidity, and wind speed without including human thermal sensation and thermal comfort, hence the recommendation of thermal comfort requirements by ASHRAE 55 [12] and ISO 7730 [16], according to [34]. All the IEQ parameters (including the noise and PM) were measured concurrently in a passenger car for each trip during the journey. The measurements were taken during the first morning trip, starting at 08.00 h, originating from Lagos to Ibadan, and during the return trips, with the departure time to Lagos at 16.00 h. The average journey time spanned two hours. Meanwhile, a few measurements performed in the waiting areas before departure at various stations have been presented because many passengers usually wait before the scheduled departure time.

Evaluation method for the ventilation parameters: The recommended indoor fresh air flow rate, Q $m^3$/h per person corresponding to achieving the recommended $CO_2$ concentrations (1000 ppm to 1200 ppm) correlations, is as follows: for 1200 ppm (2160 mg/$m^3$)

of $CO_2$ concentration, Q should be 24 m$^3$/h, while for 1000 ppm (1800 mg/m$^3$), Q should be 30 m$^3$/h per person. An illustration using the first train trip scenario has first been presented whereas all the ventilation parameters for all the trips were also calculated as shown in the Results and Discussion Section. Using Equations (1) and (2) and the average $CO_2$ level of 450 ppm (810 mg/m$^3$) for $C_{external}$, and the average $CO_2$ values to compute the $C_{equilibrum}$ in mg/m$^3$, the calculations were made. Also, it was observed during the investigations that all cabins were always fully occupied, a passenger count (PC) of 80 persons were seated for more than 90% of the assessed travel time.

$$Q = (G) / \left( C_{equilibrum} - C_{external} \right) \tag{1}$$

$$\lambda_v = \frac{Q}{V} = \frac{1}{t} \tag{2}$$

Evaluation method for the noise parameters: These noise sources are typically both airborne and structural. In the current study, only the noise levels have been assessed during six of the fifteen trips investigated. The noise source was not investigated nor categorized. The current study has evaluated the noise levels using the noise equivalent level ($L_{eq}$), a descriptor used to characterize the sound effect on humans in the evaluation of noise in vehicles. The $L_{eq}$ represents the sound pressure level (SPL) of a continuous constant sound that would have produced the same sound energy at the same time as the actual noise history [53]. Meanwhile, the raw data of the noise measurements correspond to the time series of the A curve weighted SPL values, with the sound meter configured in Fast Mode (125 ms, sampling time), as typically used to evaluate automotive noise. The $L_{eq}$ values were calculated for a time interval of the travel duration in each of the investigated train trips [12], and it can be quantitatively determined by using Equation (3).

$$L_{eq} = 10 \log_{10} \sum_{i=1}^{n} \left\{ \left( 10^{\frac{SPL_i}{10}} \times t_i \right) \right\} \tag{3}$$

$L_{eq}$ (dBA) is the noise equivalent level for each trip, $SPL_i$ is the sound pressure level measured in dBA, n is the total number of samples taken on each trip, and $t_i$ is the fraction of the total sample time. The study results are now presented graphically, analyzed, and discussed in the following sections accordingly.

## 3. Result and Discussion

This study objectively assesses the indoor environmental quality (IEQ) conditions that train passengers are exposed to in Nigeria, as the region prospects a revamp and expansion of railway transportation. This study offers scientific data on the IEQ of a transport microenvironment (trains), an exemplary developing tropic, with a paucity of scientific studies, and the need for reforms in IEQ development, infrastructure, and policy. Ultimately, it highlights gaps and recommendations for future investigations and possible stakeholder interventions. The results are presented in the following subsections for thermal comfort, indoor air quality, and other comfort parameters such as lighting and noise. The indoor climate, considering thermal comfort and the indoor air quality, was assessed for two seasons, while the particulate matter and noise parameters were reported for the six trips in the dry season. Table 2 presents the mean and standard deviation of all the investigated IEQ parameters.

**Table 2.** Mean and standard deviation of all IEQ parameters measured during all train trips.

| Trips | $T_O$ (°C) | SD | RH (%) | SD | Light (lux) | SD | $CO_2$ (ppm) | SD | VOC index | SD |
|---|---|---|---|---|---|---|---|---|---|---|
| 1 | 23.6 | 0.7 | 36 | 2.3 | 48 | 27 | 820 | 133 | 12 | 4.7 |
| 2 | 24.2 | 0.8 | 53 | 3.1 | 53 | 15 | 924 | 120 | 47 | 6.0 |
| 3 | 25.8 | 0.9 | 36 | 5.3 | 77 | 53 | 789 | 60 | 13 | 10.2 |
| 4 | 22.9 | 1.0 | 49 | 8.7 | 1556 | 1308 | 696 | 115 | 38 | 17.1 |
| 5 | 22.9 | 0.5 | 39 | 2.4 | 863 | 460 | 894 | 111 | 18 | 4.8 |
| 6 | 23.3 | 0.4 | 59 | 2.3 | 628 | 1423 | 1298 | 90 | 58 | 4.6 |
| 7 | 21.3 | 1.1 | 58 | 8.4 | 30 | 10 | 1368 | 84 | 56 | 16.5 |
| 8 | 24.6 | 1.1 | 41 | 2.2 | 47 | 31 | 1242 | 174 | 22 | 4.3 |
| 9 | 22.9 | 1.0 | 54 | 4.9 | 42 | 15 | 1684 | 151 | 48 | 9.6 |
| 10 | 20.0 | 0.2 | 51 | 1.0 | 74 | 33 | 1288 | 30 | 42 | 1.9 |
| 11 | 21.2 | 1.1 | 47 | 0.9 | 128 | 245 | 1024 | 41 | 33 | 1.8 |
| 12 | 21.9 | 1.1 | 47 | 11.2 | 168 | 160 | 1787 | 289 | 34 | 22.0 |
| 13 | 21.4 | 0.9 | 54 | 6.6 | 76 | 27 | 1493 | 203 | 48 | 13.1 |
| 14 | 20.8 | 0.7 | 52 | 3.2 | 245 | 101 | 1451 | 127 | 43 | 6.3 |
| 15 | 25.3 | 1.0 | 49 | 8.1 | 29 | 20 | 1170 | 100 | 37 | 15.9 |

Thermal comfort parameters: Ensuring adequate thermal comfort can enhance the perceptions of IAQ [54] besides its significant impact on the overall passenger comfort in the indoor transport environment. The investigated train cabins were equipped with curtains that minimized the infiltration of solar radiation in the cabins. However, the passengers sitting by these windows determined how these curtains were used, open or partially closed. Meanwhile, the current study scope did not include the assessment of thermal radiation. The studied trains were air-conditioned and during all trips, the coaches assessed all 80 seats, which were always all occupied in the selected cabins for the trips investigated. The omnidirectional IEQ multiprobe device measured five parameters including the operative temperature and relative humidity, and the graphical representations of these parameters are presented and analyzed accordingly to categorize the thermal environments in the train cabins investigated.

Figure 3 shows the average and standard deviation values of the operative temperature and relative humidity for all the trips, using as a background a color scheme with the thermal comfort quality categories, for the cooling season, of the EN16981-1 standard [48]. As shown in Figure 3a, the mean $T_O$ (°C) values of trips 7, 10, and 14 were in the discomfort zone according to the C requirements. The thermal environment was of lower temperatures (less than 21.5 °C), perhaps due to overcompensations from the HVAC settings, which also impacted the PPD for these trips. Given that outdoor temperatures in the south-western region of Nigeria, such as in the investigated case study area, are sometimes akin to those levels recorded during the summer for some countries including Spain and Portugal with Mediterranean and temperate climates, we may compare the summer indoor thermal expectations of these regions to those in the current study. Also, there are limited studies including the significant absence of well-defined local IEQ standards in the study region (Nigeria), hence the reference to other known standards like the EN16981-1 [48]. Nine of the fifteen trips exceeded the suggested temperature upper limit of 21.8 °C by Nicol et al. [55], which he prescribed in agreement with the corrected effective temperature (CET) limit for comfort proposed by Bell et al. [56], applicable to sedentary people during summer conditions whereas none of the computed $T_O$ averages exceeded the recommended physiological and safety temperature limit of 30.6° [56]. As shown in Figure 3b, the RH (%) was adequate between 35% and 65%, within the prescribed safe and comfort limit of 40% to 60% [57]. The RH (%) values computed for this study followed the EN 13129-1 [46] requirements: RH < 65% when T ≤ 23 °C, and RH < 45% when T ≤ 29 °C [58]. Also, Božič et al. recommended an approximate range of 50% for the minimal risk and spread of infectious aerosols in indoor spaces [59]. The mean values of the current study rolling stock did not exceed the prescribed values for a guaranteed pleasant interior, considering

the recommended values (in the EN 13129-1 [46], mainline rolling stock) of 27 °C and 51.6% maximum mean interior temperature and RH, respectively, in summer conditions for southern European countries (zone 1), which have outdoor temperatures nearing 40 °C [47].

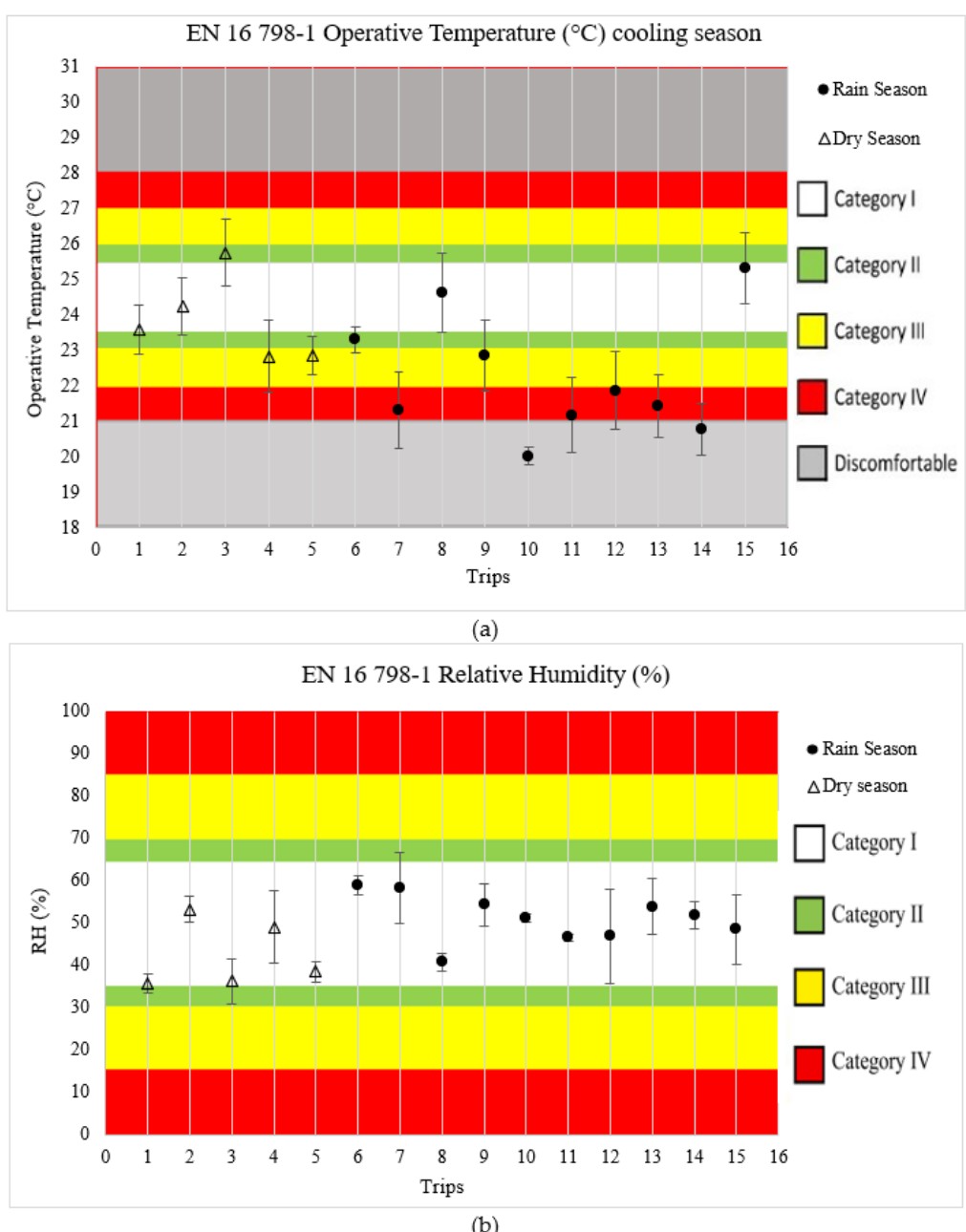

**Figure 3.** Mean and SD values of thermal comfort parameters: (**a**) the $T_O$ (°C), and (**b**) the RH (%) in all trips investigated.

PMV (predicted mean vote) and PPD (pre-visible percentage of dissatisfied) index: The thermal comfort sensation of the exposed occupants in the studied train is quantitatively predicted using the PMV–PPD index now presented and analyzed accordingly. Table 3 presents the PMV–PPD indices computed from the averages of the thermal comfort parameters considering a metabolic rate of 1 met and 0.6 clo has been adopted, considering that for both seasons, the clothing worn is typically summer-like and the HVAC is mostly for cooling in this tropic region.

**Table 3.** PMV–PPD parameters for all trips.

| Trips | PMV | PPD | Season | HVAC Use |
|---|---|---|---|---|
| 1 | −0.53 | 10.80 | Rain | Cooling |
| 2 | −0.31 | 7.00 | Rain | Cooling |
| 3 | 0.26 | 6.40 | Rain | Cooling |
| 4 | −0.77 | 15.50 | Rain | Cooling |
| 5 | −0.77 | 17.60 | Rain | Cooling |
| 6 | −0.63 | 13.40 | Dry | Cooling |
| 7 | −1.34 | 42.20 | Dry | Cooling |
| 8 | −0.17 | 5.60 | Dry | Cooling |
| 9 | −0.77 | 17.60 | Dry | Cooling |
| 10 | −1.79 | 66.00 | Dry | Cooling |
| 11 | −1.37 | 44.00 | Dry | Cooling |
| 12 | −1.13 | 31.80 | Dry | Cooling |
| 13 | −1.3 | 40.30 | Dry | Cooling |
| 14 | −1.51 | 51.50 | Dry | Cooling |
| 15 | 0.08 | 5.10 | Dry | Cooling |

Figure 4 shows the PMV values computed for all the trips. Six trips are observed in the discomfort zone and category IV. On most trips, the thermal environments were not adequate according to the ASHRAE and EN 16798-1 requirements. It was observed during most trips that the air conditioning made the cabins rather cold, as shown by the PMV index. The HVAC systems may have been set to overcompensate for the typical high outdoor temperature in tropical regions. Also, most people in the tropics are dressed in summer-like clothing and coupled with the low metabolic rate of sedentary persons, the thermal sensation accounts for higher PPD values. Also, following the PMV index graph, higher PPD values were computed for the rainy season trips. In the dry season, outdoor temperatures are usually higher than in the rainy season; therefore, the AC settings should be regulated accordingly. Otherwise, the same settings may account for colder cabins since during both seasons in the tropics, AC works for cooling. Most of the rainy season trips recorded a higher PPD, suggesting that adequate regulation of cabin AC is needed to avoid overcompensation in cabin cooling. Summarily, thermal comfort still requires some intervention in some of the assessed train cabins, perhaps better HVAC settings to achieve prescribed limits while the adequate use of curtains for shading can reduce overall thermal loads vis-a-vis energy used for climatization.

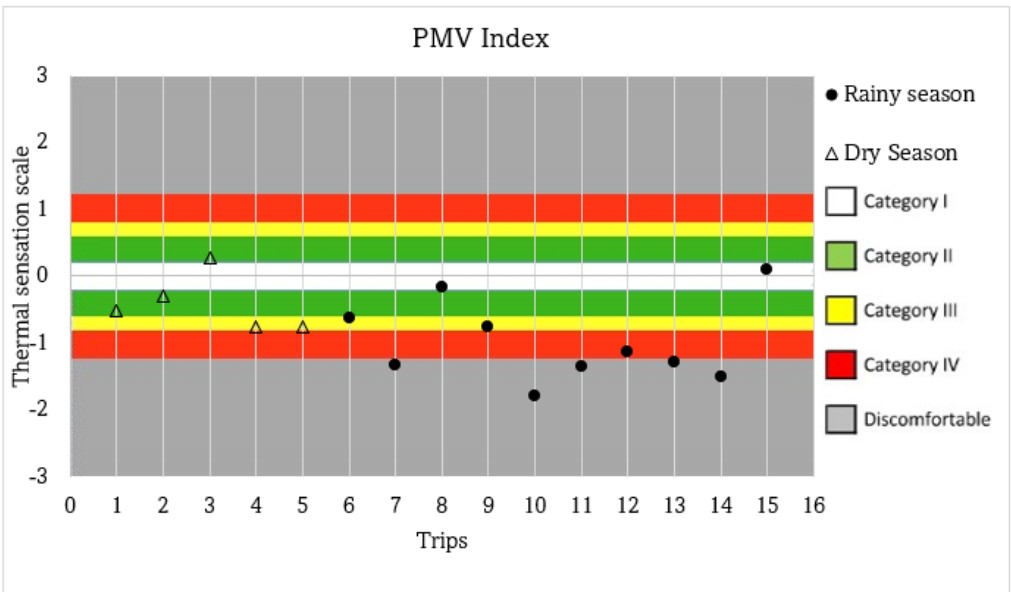

**Figure 4.** Graphical representation of the PMV index for all train trips investigated.

Indoor air quality parameters: The indoor concentrations of PM, inorganic compounds ($CO_2$, CO, $O_3$, $SO_2$, $NO_X$), and organic compounds (benzene, polyaromatic hydrocarbons, VOCs), including biological organisms (fungi, viruses, bacteria), are common determinants of the IAQ of transport indoor microenvironments. There are risks to health, comfort, and general safety to exposed occupants if acceptable IAQ levels are violated. PM concentrations exceeding the threshold are of significant concern because studies have reported negative health outcomes associated with PM penetrating respiratory and pulmonary systems depending on their size [60]. Also, short, or chronic exposure to $CO_2$ exceedance levels have been associated with a myriad of health risks, decreased cognition and performance, and increased discomfort in exposed occupants [61]. The present study's results, regarding the investigated IAQ parameters ($CO_2$, VOCs, $NO_2$, $PM_{2.5}$, and $PM_{10}$) during the train trips, are presented below. The averages and standard deviation values have been categorized according to the EN 16 798-1 requirements.

Figure 5 shows the mean and standard deviation distribution of the $CO_2$ and VOC parameters as measured using the IEQ multiprobe device, whereas these measured values have been categorized according to EN16798-1 and evaluated [45]. In Figure 5a, the $CO_2$ levels (ppm) fall into categories I, II, III, and IV. There was no trip in the discomfort zone. Comparatively, trip 12 presents the poorest indoor climate parameters considering the mean $CO_2$ level (1786.80 ppm in category IV) and the PPD (31.8%) reported. Meanwhile, the average VOC levels computed suggest that the VOCs were within acceptable limits for all trips investigated (Figure 5b). Although no subjective evaluations concerning the IAQ and perceived air quality (PAQ) were performed for the current study, it is possible to deduce from the results of the experimental value averages obtained for the $CO_2$ and VOCs that the IAQ was acceptable in the cabins investigated considering the categorization and requirement of the EN16798-1 standard. Regarding the PM, the TROTEC BQ400 particle counter and FLOW air quality devices were used to investigate six trips. The PM average and standard deviation values are presented in Table 4 as computed from the measurements obtained from the two particle counter devices and analyzed by the known standard limits. Martins et al., in a review study, reported that no PM regulatory standards were found for most African countries and Asian countries [60]; therefore, the current study's results were evaluated according to the acceptable requirements of standard regulations from other regions.

**Table 4.** Fresh air flow rate and air exchange rate parameters in all trips.

| Trip | PC | $C_{ext}$ (mg/m$^3$) | G (mg/m$^3$) | $C_{equi}$ (mg/m$^3$) | Q (mg/h$^3$) | Q (all PC) | PCV (m$^3$) | $\lambda$ (h$^{-1}$) |
|---|---|---|---|---|---|---|---|---|
| 1 | 80 | 810 | 37,000 | 1476 | 55.6 | 4444.4 | 145.6 | 30.5 |
| 2 | 80 | 810 | 37,000 | 1667 | 43.2 | 3452.7 | 145.6 | 23.7 |
| 3 | 80 | 810 | 37,000 | 1424 | 60.3 | 4823.2 | 145.6 | 33.1 |
| 4 | 80 | 810 | 37,000 | 1256 | 83.0 | 6638.3 | 145.6 | 45.6 |
| 5 | 80 | 810 | 37,000 | 1613 | 46.1 | 3685.7 | 145.6 | 25.3 |
| 6 | 80 | 810 | 37,000 | 2342 | 24.1 | 1931.9 | 145.6 | 13.3 |
| 7 | 80 | 810 | 37,000 | 2468 | 22.3 | 1784.9 | 145.6 | 12.3 |
| 8 | 80 | 810 | 37,000 | 2241 | 25.9 | 2068.3 | 145.6 | 14.2 |
| 9 | 80 | 810 | 37,000 | 2941 | 17.4 | 1388.9 | 145.6 | 9.5 |
| 10 | 80 | 810 | 37,000 | 2318 | 24.5 | 1962.3 | 145.6 | 13.5 |
| 11 | 80 | 810 | 37,000 | 1843 | 35.8 | 2864.9 | 145.6 | 19.7 |
| 12 | 80 | 810 | 37,000 | 3217 | 15.4 | 1230.0 | 145.6 | 8.4 |
| 13 | 80 | 810 | 37,000 | 2687 | 19.7 | 1576.6 | 145.6 | 10.8 |
| 14 | 80 | 810 | 37,000 | 2612 | 20.5 | 1642.8 | 145.6 | 11.3 |
| 15 | 80 | 810 | 37,000 | 2106 | 28.5 | 2284.0 | 145.6 | 15.7 |

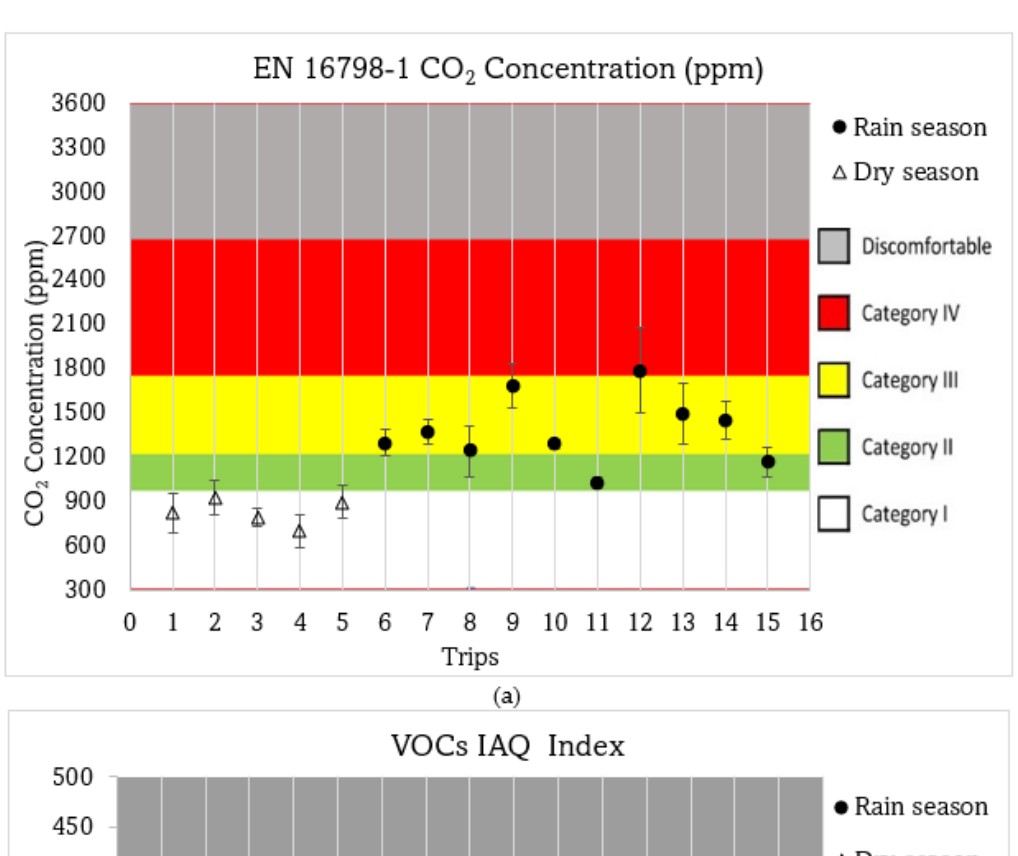

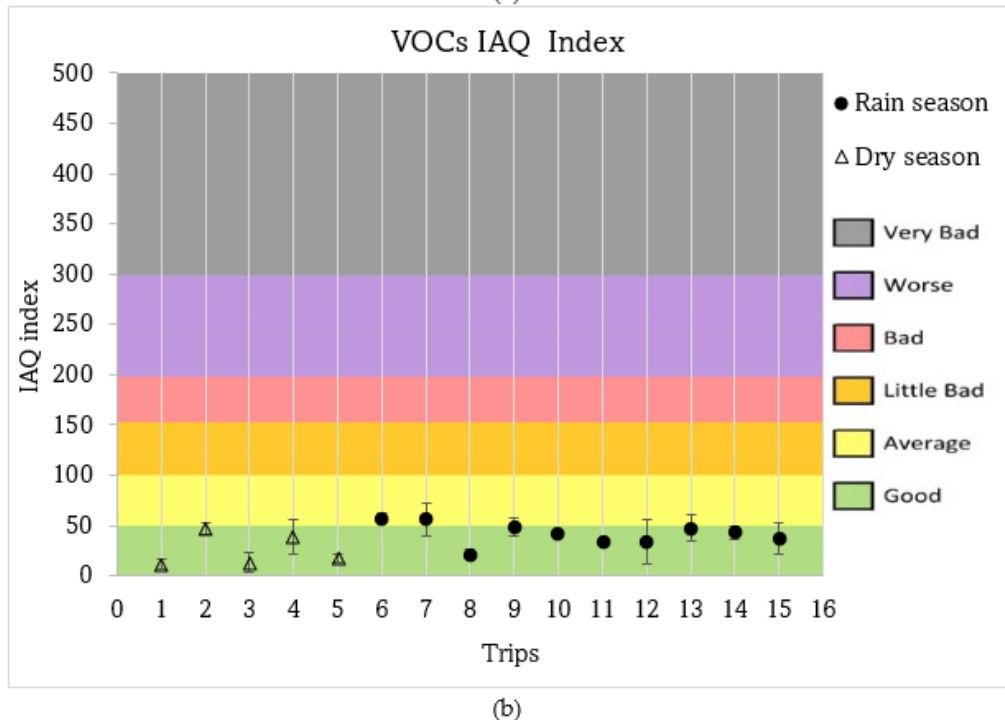

**Figure 5.** Mean and SD values of IAQ parameters in all train trips: (**a**) the $CO_2$ levels and (**b**) shows the VOC levels.

Ventilation parameters: The fresh air flow rate, Q (m³/h), and air exchange rate, λ (h⁻¹), are essential ventilation parameters that impact the IAQ of indoor spaces. The results of the calculated parameters using the equations are presented in Table 4.

The calculation illustration using the first train trip is as follows: PC of 80 persons (fully occupied cabins in all seating without standing). With an internal width of 2.8 m, interior height of 2.0 m, and a coach length of 25.9 m per coach, we can obtain the passenger car volume (PCV).

$$PCV = L \times W \times h = 145.6 \text{ m}^3$$

Taking $C_{external}$ to be 810 mg/m$^3$ (equivalent to an outdoor $CO_2$ of 450 ppm), $C_{equilibrum}$ is 1476 mg/m$^3$ (820 ppm), and the generated $CO_2$ per occupant, G as 37,000 mg/m$^3$, the Q = 55.6 m$^3$/h per passenger. A fresh air flow rate of 55.6 m$^3$/h implies that the occupants of the illustrated trip's coach were exposed to an overventilated cabin. The results presented in Table 4 show that, in 9 of the 15 trips, the fresh air flow rates Q (mg/h$^3$) were inadequate, while the remaining trips were within the recommended range (24 m$^3$/h to 30 m$^3$/h). Meanwhile, the fresh flow rate of the fully occupied coach was 4444.4 m$^3$/h (the Q per person multiplied by the PC). Using Equation (2), the air exchange rate was calculated as follows.

$$\lambda_v = \frac{Q}{V} = \frac{444.4}{145.6} = 30.5\ h^{-1}$$

Regarding the air exchange rate, $\lambda_v$ (h$^{-1}$), to achieve the $CO_2$ concentration of a range of 1000 ppm to 1200 ppm for an ideal IAQ in the coaches (taking PCV as 145.6 m$^3$), a range of 13 h$^{-1}$ to 16.5 h$^{-1}$ is required. Comparing the air exchange rates obtained (Table 4) to this requirement, only four trips were within the ideal range of 13 h$^{-1}$ and 16.5 h$^{-1}$, although two trips, 7 and 9, were near suitable in the air exchange rate and their fresh air flow rate parameter values, 22.3 mg/h$^3$ and 25.9 mg/h$^3$. In total, 60% of the studied train coaches have shown inadequate levels of fresh air flow rate and air exchange rate parameters, suggesting that ventilation gaps exist and the need for IAQ interventions.

Particulate matter: Assessing the in-cabin PM level in addition to $CO_2$ is also crucial to defining the IAQ condition because it is an air pollutant that can result in a myriad of health problems in exposed occupants to unrecommended thresholds. $PM_{2.5}$ (all the airborne particles not exceeding 2.5 μm in diameter) is an inhalable fine particulate that can pose a serious risk to human health depending on its chemical composition whereas $PM_{10}$ (all the airborne particles not exceeding 10 μm) is also inhalable and can be harmful to human health. Table 5 presents the mean and SD of the measured parameters ($PM_{2.5}$, $PM_{10}$, $NO_2$, and VOCs) for 6 of the 15 studied trips.

**Table 5.** Computed PM, $NO_2$, and VOC measurements from the two devices.

| Trips | TROTEC BQ400 Device | | | | FLOW Air Quality Device | | | | | | | |
|---|---|---|---|---|---|---|---|---|---|---|---|---|
| | $PM_{2.5}$ (μg/m$^3$) | | $PM_{10}$ (μg/m$^3$) | | $PM_{2.5}$ (μg/m$^3$) | | $PM_{10}$ (μg/m$^3$) | | $NO_2$ (ppb) | | VOC (ppb) | |
| | Mean | SD | Mean | SD | Mean | SD | Mean | SD | Mean | SD | Mean | SD |
| 1 | 51.3 | 14.7 | 99.4 | 14.3 | 45 | 7 | 115 | 15 | 40 | 19 | 87 | 31 |
| 2 | 49.5 | 19.1 | 99.3 | 17.2 | 43 | 11 | 108 | 22 | 16 | 4 | 113 | 44 |
| 3 | 24.4 | 7.9 | 54.9 | 17.5 | 29 | 10 | 108 | 13 | 0 | 0 | 159 | 18 |
| 4 | 25.9 | 6.6 | 72.1 | 15.8 | 32 | 9 | 109 | 20 | 6 | 3 | 190 | 16 |
| 5 | 51.3 | 30.4 | 79.3 | 40.3 | 34 | 7 | 87 | 21 | 58 | 4 | 117 | 32 |
| 6 | 22.5 | 7.9 | 47.9 | 21.2 | 28 | 5 | 109 | 10 | 20 | 18 | 146 | 27 |

PM was assessed in the dry harmattan season, and the mean values computed from the FLOW air quality device were comparable to the values computed for the TROTEC BQ 400 device. The mean PM levels computed show exceedances, referencing the EU outdoor limits, comparable to the evaluations performed by Maggos et al. for the in-train measurements of PM and $NO_2$ levels exceeding the outdoor daily limits for $PM_{10}$ (50 μg/m$^3$), $PM_{2.5}$ (25 μg/m$^3$), and hourly limit for $NO_2$ (200 μg/m$^3$). The indicative comparison of poor IAQ, as described by Maggos et al. [62], was due to the infiltration of smokestack emissions into the investigated cabins. Rivas et al. [63], assessing commuter exposure to air pollution in different transport modes and routes including other varied influences, reported higher PM concentrations in underground trains with openable windows as $PM_{2.5}$ = 37.4 μg/m$^3$, while non-openable window trains, as those of the current study were, as $PM_{2.5}$ = 16.4 μg/m$^3$ (non-open windows), which is less than the mean $PM_{2.5}$ values reported for all the trips of the current study trains traveling at the ground surface level.

Figure 6 presents a graphical distribution of the mean and standard deviation values computed for all the trips using the two devices. The similarity in the graphical distribution of the computed mean and deviation PM values considering both counting devices indicates that the in-train PM levels were in exceedance of the recommended limits by the WHO, ASHRAE, and the Portuguese legislation ($PM_{2.5}$ limit of 25 $\mu g/m^3$ and $PM_{10}$ limit of 50 $\mu g/m^3$) in all the investigated train trips. The $PM_{10}$ levels measured (TROTEC device: 47.9 $\mu g/m^3$–99.4 $\mu g/m^3$, and Flow device: 87 $\mu g/m^3$–115 $\mu g/m^3$) were comparatively higher for all the trips than $PM_{2.5}$. (TROTEC device: 22.5 $\mu g/m^3$–51.3 $\mu g/m^3$, FLOW device: 28 $\mu g/m^3$–45 $\mu g/m^3$). Contrastingly, the VOCs were not in exceedance in the current study unlike the findings by Russi [30] that the VOCs (2516 $\mu g/m^3$) were significantly high in trains. Similarly, the VOC levels in the current study were found to be lacking, as reported by Ongwandee et al. [24]. Considering ASHRAE's recommendation that in-cabin $CO_2$ should not exceed 1000 ppm [64], only five computed mean $CO_2$ values, mostly the trips surveyed in the dry season, complied. However, no mean $CO_2$ value computed was of toxic concentration levels. Also, all the mean PM values computed were less than the permissible limit of 150 $\mu g/m^3$ in contrast to the findings of Masyita et al. [23]. The preliminary assessments of the IAQ indicate that its inadequacy was not toxic, although the elevated PM levels and typical risk of high occupancy density in these regions are good reasons to ensure adequate HVAC settings to enhance fresh air and indoor climate conditions.

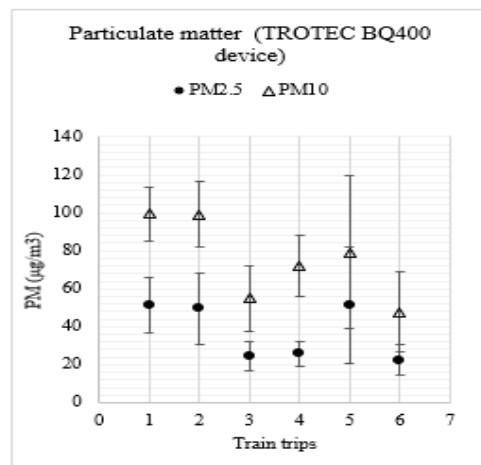 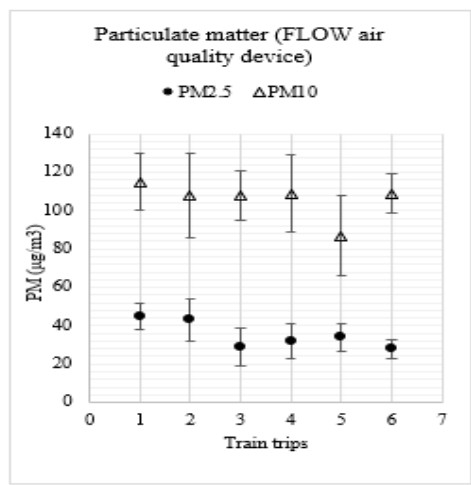

**Figure 6.** The mean and SD values of PM by the two measuring devices.

The indoor climate in passenger waiting areas: The indoor climate of some passenger waiting areas has also been investigated and reported, categorizing and analyzing the exposure according to the EN 16798-1 requirements. Typically, passengers, including railway workers, are exposed to the indoor climate of the waiting areas, which sometimes accommodates the ticket service areas. Due to the train schedules and apprehension to avoid missing the trains, passengers sometimes usually wait for varying lengths of time until departure. The indoor climate, including the thermal comfort and IAQ parameters, was within permissible limits considering the temperature, $CO_2$, RH, and VOCs. The investigated waiting areas have been recently built and equipped with functional HVAC, while the frequently open entrance and exit doors allow for the easy infiltration of fresh outdoor air. Table 6 presents the mean and SD values of the IEQ parameters measured by the IEQ multiprobe device. The values computed indicate a measure of compliance with the prescribed indoor climate standard requirements.

**Table 6.** Mean and standard deviation value IEQ parameters of passenger waiting areas.

| Station | $T_O$ °C | SD | RH (%) | SD | $CO_2$ (ppm) | SD | VOC Index | SD |
|---------|------|-----|--------|-----|--------------|-----|-----------|-----|
| AG | 21.9 | 0.1 | 57 | 1.6 | 631 | 41 | 53 | 3.1 |
| MJ | 27.4 | 0.4 | 69 | 1.8 | 541 | 35 | 77 | 3.4 |
| OA | 24.2 | 0.9 | 46 | 2.8 | 627 | 22 | 32 | 5.5 |
| OA | 23.0 | 0.4 | 60 | 0.8 | 801 | 21 | 59 | 1.6 |

Figure 7 shows the main indoor climate parameters. In Figure 7b,d, the mean $CO_2$ (ppm) and VOCs computed suggest that the IAQ was adequate. Also, the mean values of TO (°C) and RH (%) (in Figure 7a,c) are indicative of good thermal environments in the assessed passenger waiting areas. The waiting areas were also huge enough to accommodate the high passenger density as observed during the time of the current study survey. However, the measurement of PM was not conducted for the passenger waiting areas due to train management permission and other logistics for the field study. It is recommended that future studies assess the infiltration of PM and VOCs, including VICs (volatile inorganic compounds), to adequately characterize the IAQ of exposed passengers as many stations are situated around unpaved surroundings, vehicle traffic-prone areas, markets, and industries.

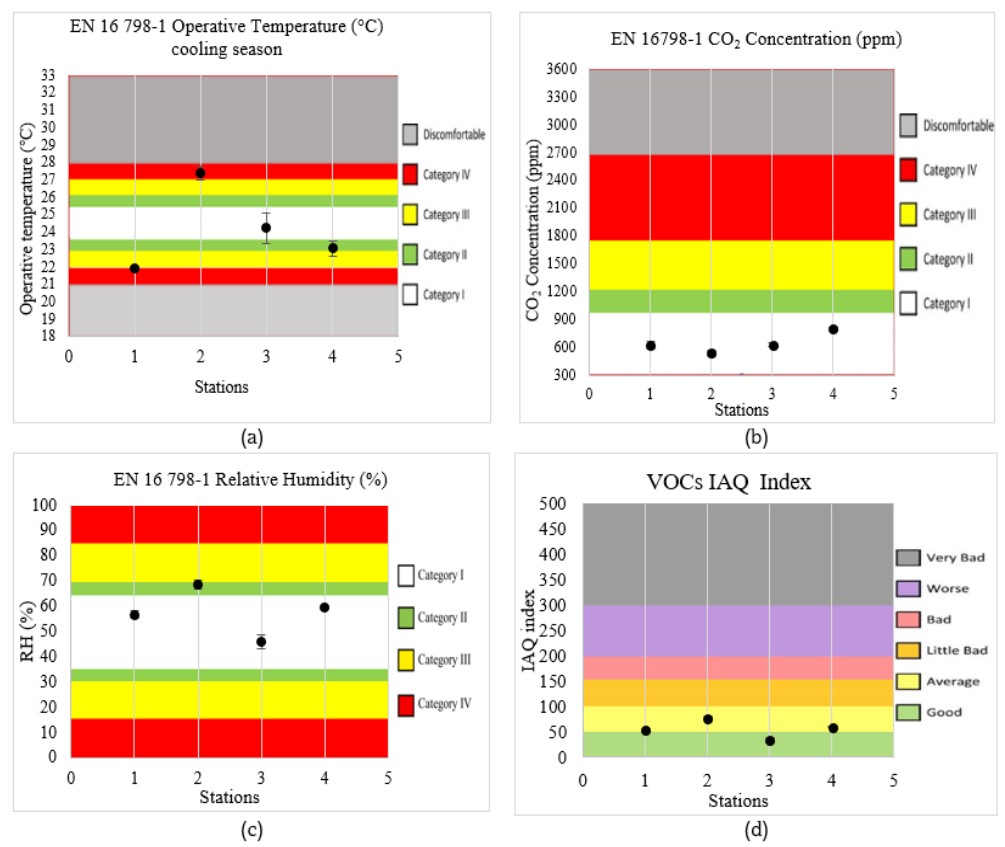

**Figure 7.** Graphical representation of the mean and SD indoor climate parameters of passenger areas: (**a**) operative temperature values, (**b**) $CO_2$ concentration level, (**c**) relative humidity, and (**d**) VOC index.

Noise: Noise is referred to as unwanted sound. In vehicle and train cabins, passengers are prone to noise exposure from internal and external sources. The noise equivalent levels for each of the assessed train trips was calculated according to Equation (3) and presented graphically. The noise levels have been analyzed according to the standard limits found in the accessible published science literature; however, there was no record of any national

indoor noise limit or regulation for mass transit vehicles nor train passenger cars for Nigeria or similar sub-Saharan African countries as at the time of the current study report.

In Figure 8, the calculated $L_{eq}$ (dBA) is presented, and the various limits are reported. The noise equivalent was computed for each trip, as shown in Figure 8a, while Figure 8b presents the comparison of the interior noise evolution and the $L_{eq}$ (dBA) in one of the studied trains (trip 4) with the accessible referenced standards now discussed. The total journey time did not exceed three hours in any of the investigated train trips; therefore, no $L_{eq}$ computed violated the Occupational Safety and Health Administration (OSHA) 85dBA 8 hr exposure limit. The highest value obtained for $L_{eq}$ was 85 dBA. According to Bryan et al.'s [65] noise criterion for passengers in vehicles, noise levels were designated as quiet if at 67 dBA, noticeable at 73 dBA, intrusive at 79 dBA, annoying at 85 dBA, and very annoying at 91 dBA. The $L_{eq}$ calculated for each trip surveyed suggests that the noise levels during trips 2, 3, and 4 were annoying while on the trip, the noise level could be simply noticeable but trips 5 and 6 are better described as quiet. However, the target is to have an A-weighted equivalent interior sound pressure level of between 65 dBA and 70 dBA, which is commonly chosen to achieve passenger comfort [66]. Recently, Peng et al. [34] presented relevant train interior noise standard limits for the China Railway Corporation (GB/T 12816 [50]), the European Union (UIC 660-2002 [67]), the Japanese Industrial Standard (JIS), and the USA (USFS). According to the UIC 660-2002 [67] noise $L_{Aeq}$ requirements for open air with train speeds of 250 km/h, for the center-of-coach it is 65 dBA, for the passing platform it is 75 dBA, and for the windshield it is 80 dBA. The $L_{Aeq}$ in the current study pertains to center-of-coach, implying that only trip 6 ($L_{Aeq}$ of 64.1 dBA) complied. Furthermore, the USFS recommends that for speeds greater than 75 km/h, in any train track and electric multiple units (EMUs), the $L_{max}$ should not exceed 93 dB. This implies that all trips in the current study followed the USFS train interior noise requirement. The GB/T 12816 [50] requirement (tunnel or open air, not specified) for a speed of 200 km/h is 78 dBA for the train cab, 65 dBA for the center-of-coach, and 67 dBA for the end-of-coach. Only trip 6 in the current study complied with the GB/T 12816 [50] requirement, considering the 65 dBA limits for the center-of-coach. The findings by Winter et al. [2] on passenger comfort sensation regarding noise-simulated different scenarios of tunnel and line conditions, involving 60 participants, indicate that occupants were more comfortable with noise levels of 62 dBA in the tunnel and line conditions than at 72 dBA. Comparatively, the lowest $L_{eq}$ computed in the current study (64.1 dBA) exceeds the comfort level of 62 dBA reported by Winter et al. Meanwhile, the Brazilian N-17 Standard requirements indicate that noise exposure exceeding 65 dBA during 8 h of a workday is considered uncomfortable [68]. Although most $L_{eq}$ computed in the current study exceeds this limit, the maximum time was less than 3 h during each trip, considering that besides the passengers, train agents (workers) and cabin hands were also occupants during all trips. Invariably, there is a need to ensure a better acoustic environment in most of the trains investigated by the current study, although the noise levels may not pose an occupational hazard considering the OSHA and Nigeria's National Environmental Standards and Regulations Enforcement Agency (NESREA) occupational noise exposure limit of 85 dBA for an 8 h working period [69]. There is no known national regulatory standard for interior noise in transport vehicles in Nigeria; most regulations concern environmental noise. Given that the trains in Nigeria are still of low- and mid-range speeds, including other peculiarities relating to transport infrastructure, climate, and policy, it is therefore suggested that an adapted national railway standard be developed for the region. It is important to mention that other sound descriptors exist that are useful in the evaluation of the acoustic environment of indoor spaces including vehicle and train microenvironments. Apart from the computed $L_{eq}$ per trip, which represents the sound pressure level (SPL) of a continuous constant sound that would have produced the same sound energy in the same time (T) span as the actual noise history, the current study has not evaluated other sound descriptors and indices such as the articulation index (AI), predicted speech interference level (PSIL), composite rate of preference (CRP), including

other sound parameters such as the sound power level (PWL), and the sound intensity level (SIL) [53]. Particularly, the evaluation of the AI, a frequency analysis of sound, and a useful index to characterize the influence of parasite noise on the intelligibility of speech [70], in indoor spaces as in transport passenger compartments could be a useful noise descriptor in developing tropics such as Nigeria where several studies [71–74] have reported significant existential noise in the outdoor and indoor spaces. The indoor noise in transport cabins includes noise intrusion from ambient outdoor and internal noise sources including the vehicle (structural and engine sources and occupant sources like talking, use of portable gadgets, and other behavioral activities). Furthermore, considering some cultural and communication peculiarities of commuters in mass transit vehicles in Nigeria, it is recommended for consideration in future studies to evaluate the AI.

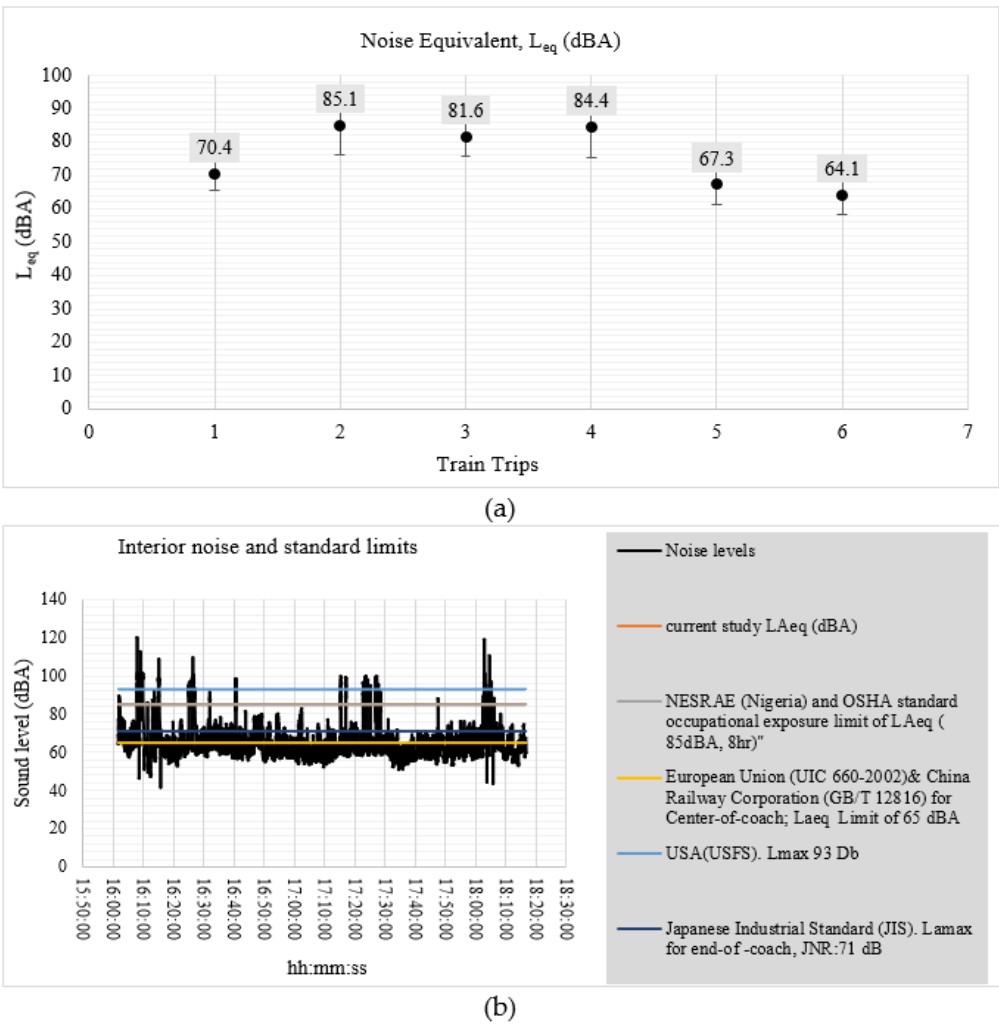

**Figure 8.** Graph showing noise equivalent $L_{eq}$ (dBA) for each train cabin assessed and interior noise for a selected train trip. (**a**) presents the $L_{eq}$ levels in each trip and (**b**) the interior noise of a selected train passenger car compared to the referenced standards.

Visual Comfort: Indoor lighting impacts occupants' comfort, well-being, and safety. Nowadays, the use of energy-efficient lighting, adequate illuminance, suitable illuminance distribution, color temperature, and seat arrangements including the interior decorative materials for better passenger comfort and ride experience in trains is crucial to assure sustainability and profitability for train operators. If trains are more comfortable for passengers, in experience and safety, there is a likelihood that patronage of trains for travel will increase. Hence, the current study has objectively assessed the cabin illuminance during all the trips investigated.

Figure 9 presents the graphical representation of the average lighting levels computed during each of the trips investigated. In Nigeria and many developing tropics, there are no guidelines for rail transit passenger compartment lighting design. EN 13272-2012 [75] recommends that the average illuminance should be 150 lux with an illuminance uniformity of 0.8 lux–1.2 lux in the passenger seat area. In standing areas, an average illuminance of 50 lux with an illuminance uniformity of 0.5 lux–2.5 lux is required. Meanwhile, other standards like JIS E 4016-2009 [76] and GB/T 7928-2003 [77] require that normal lighting areas should be 200 lux or more [78]. Following the average illuminance values (already presented in Table 2), only five passenger compartments exceeded 150 lux considering the average illuminance recorded during each trip, meeting the EN 13272-2012 [75] recommendations. In the current study, the illuminance levels were measured only for the sitting areas. Other lighting parameters such as glare and luminance were not assessed in the current study. It is important to highlight that most of the trains were equipped with window curtains that may have contributed to shading in the passenger compartment. Shading the passenger compartments could be advantageous for ensuring an adequate thermal environment (LESS heat penetration into the cabin from irradiation) in the tropics such as Nigeria; therefore, adequate lighting can be achieved in the train by equipping OLED (organic light-emitting diodes) luminaires. By visual observation, we remark that the trains investigated in the current study were not equipped with OLED luminaires; meanwhile, OLED luminaires are preferred for performance, as suggested by the studies [2,79,80].

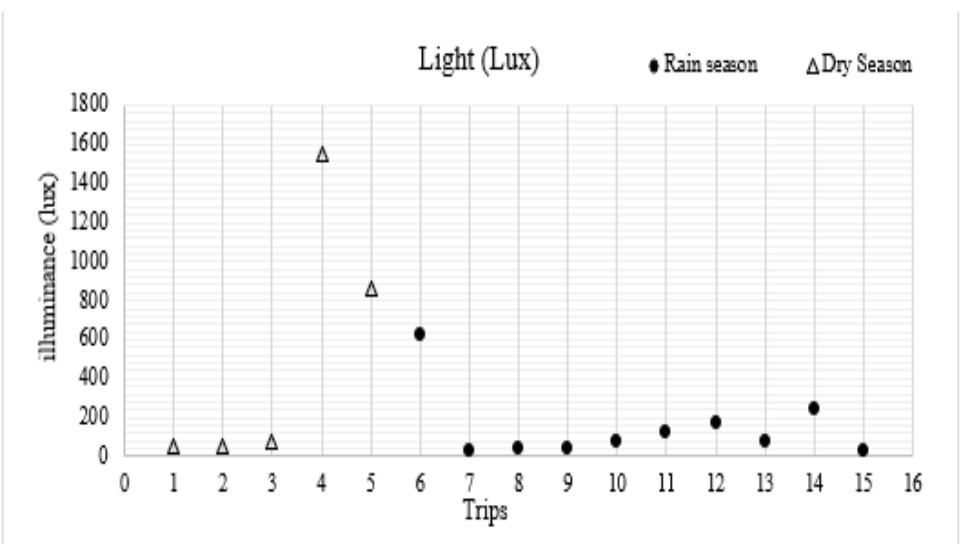

**Figure 9.** Mean illuminance and atmospheric pressure in all trips investigated.

## 4. Limitations of the Study

The current study has used only objective methods to assess and evaluate the IEQ parameters. It is recommended that for a more holistic evaluation, subjective assessments also be performed in future works. Also, other comfort parameters impacting the overall IEQ conditions such as vibration and passenger seat comfort have not been assessed in the current study. The PM measurements in the current study have been performed considering only the dry season; it could be useful to conduct a comparison of PM for the dry and rainy seasons. Furthermore, more attention and evaluation should be given to thermal comfort measurements, visual comfort technologies, and distribution in the cabin vis-a-vis the evaluation of passenger comfort. Finally, due to permit constraints, limited transport stakeholder cooperation, and the inherent challenges with real-time travel assessments, a simple methodology was adopted and adapted to focus more on the indoor climate parameters. Future works should ensure wider stakeholder collaboration and support for ensuring the use of more robust study methods and measuring tools according to the EN 13272-2012 [75] recommendations. Although this study reported that the trains were

equipped with curtains for shading, future works must assess the risk of increased thermal loads linked to high solar radiation in the tropics, which can contribute to awareness regarding thermal management and energy-efficiency targets for in-cabin climatization.

## 5. Conclusions

The current study has assessed the IEQ of passenger compartments in air-conditioned trains during real-time travel between the Lagos and Ibadan metropolitan cities in Nigeria including a preliminary assessment of the indoor climate parameters of passenger waiting areas in train stations. This study objectively assessed and analyzed the IAQ (specifically, $CO_2$, VOCs, $NO_2$, $PM_{2.5}$, and $PM_{10}$), thermal, visual, and noise comfort parameters according to the recommended limits. The indoor climate parameters were mainly categorized and analyzed according to EN 16798-1. Other IEQ parameters have also been analyzed according to some relevant requirements from the ASHRAE, EN 13272-2012, OSHA, WHO, and other known national regulations.

The main findings indicate that IEQ gaps exist concerning the passenger cabins of the studied trains. The PMV index suggests cooling over-compensation from settings in the air conditioners. Concerning the IAQ, the mean levels of the measured $CO_2$, $NO_2$, and VOC pollutants did not present toxic or severe discomfort levels, but $PM_{2.5}$ and $PM_{10}$ were in exceedance of the referenced WHO limits in six of the train cabins investigated, indicative of a poor IAQ. The current study did not characterize the source of elevated levels of PM nor its effects on exposed occupants; however, owing to previous study reports, in-cabin PM infiltration has been linked to intrusion during the opening of doors, windows, fresh air supply, ineffective filters, compromised ventilation systems, and particle resuspension phenomenon. These apply to the study case since there is elevated pollution in ambient outdoor air due to high vehicle traffic, industries, socioeconomic activities, and environmental policy gaps in the region. Also, the ventilation parameters, fresh air flow rate and air exchange rate, were inadequate in 60% of the studied train passenger cars. The findings suggest that the indoor climate of passenger waiting areas was thermally comfortable and of a non-toxic IAQ. The noise equivalent calculated in six trips assessed suggests that passengers and train workers are not exposed to annoying noise, but it is noticeable and intrusive. Although the least noise equivalent level ($L_{eq}$ of 64.1 dBA) exceeds the comfort level of 62 dBA reported by Winter et al., there was no exceedance of the exposure noise limits of the OSHA and NESREA. The mean illuminance computed suggests that only five of the fifteen trips evaluated complied with the EN 13272-2012 recommendation of 150 lx in the train passenger compartment.

The current study leverages the urgent need to improve mass transit in developing sub-Saharan countries, considering their geometric population growth, urbanization, and sustainability trends. It behooves transport stakeholders to improve passenger comfort and safety and minimize the risk to health while providing energy-efficient and sustainable mobility solutions. This study serves as the first scientific evaluation of IEQ in trains in Nigeria and the developing West African region where several efforts have recently been ongoing to revive and improve the train transportation systems. Its findings present a preliminary assessment that will help sensitize transport stakeholders to IEQ gaps. Enhancing local IEQ regulations for transport indoor microenvironments is necessary given the climate peculiarity, transport technology, culture, and high commuter traffic tendencies in developing tropics, hence the recommendation for more IEQ and outdoor environmental scientific studies and an increase in IEQ awareness among commuters and transport stakeholders in Nigeria and similar sub-Saharan countries.

**Supplementary Materials:** The following supporting information can be downloaded at: https://www.mdpi.com/article/10.3390/su152316533/s1, Table S1. The field measurement equipment parameters.

**Author Contributions:** Conceptualization, J.O.O.; methodology, J.O.O.; software, M.G.d.S.; validation, J.O.O. and M.G.d.S.; formal analysis, J.O.O.; investigation, J.O.O.; resources, J.O.O.; data curation, J.O.O.; writing—original draft preparation, J.O.O. and J.-P.K.B.N.; writing—review and

editing, J.O.O., J.-P.K.B.N., M.G.d.S. and A.S.N.R.; supervision, M.G.d.S. and A.S.N.R.; project admin-istration, M.G.d.S. and A.S.N.R.; funding acquisition, M.G.d.S. and A.S.N.R. All authors have read and agreed to the published version of the manuscript.

**Funding:** Funded by the Portuguese Foundation for Science and Technology (FCT), Portugal, under the project grant scholarship REF: UI/BD/152067/2021 and was supported by the Asso-ciated Labora-tory of Energy Transports and Aeronautics Projects FCT/UIDB/50022/2020 and FCT/UIDP/50022/2020.

**Data Availability Statement:** All data generated and analyzed during this study can be found within the published article and its Supplementary Files. They are also available from the corresponding author upon reasonable request.

**Acknowledgments:** The present work was developed in the framework of the Energy for Sustain-ability Initiative of the University of Coimbra. Also, the authors recognize the field support of H.B. Sulola.

**Conflicts of Interest:** The authors declared no potential conflict of interest concerning this article's research, authorship, and/or publication.

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
