# Peer review of "Indoor Environmental Quality Assessment of Train Cabins and Passenger Waiting Areas: A Case Study of Nigeria"

_sustainability, doi:10.3390/su152316533_

Round 1

Reviewer 1 Report

Comments and Suggestions for Authors

The paper “Indoor Environmental Quality Assessment of Train Cabins and Passenger Waiting Areas. A case study of Nigeria” presents the results obtained from field measurements of indoor air quality, thermal, acoustic, and visual environments in train passenger cabins and waiting areas, during 15 trips in the dry and rainy season of Nigeria. My observations are:

 - (Line 28): Practical implications should be included in the discussion section of the manuscript.

 - (Line 67): “Literature review:” Authors should modify the manuscript’s structure and subsections should be included.

 - (Line 178) :”… nations into Niger [39], [40].” Authors should check the reference style through the manuscript.

 - (Line 180). The quality of Figure 1 should be improved.

 - (Line 216). The characteristics of the sensors used during the field measurement campaign should be include in a new Table. The accuracy, measurement range and resolution of each sensor should be also indicated.

 - The sampling period for each of the different collected variables should also be indicated in the manuscript.

 - (Line 231). How operative temperature was determined in this study?  Were radiant temperature and air temperature measured separately? The authors should clarify these issues.

 - (Line 253). “… the OSHA and NIOSH requirements Furthermore, …” Authors should thoroughly revise the writing style of the manuscript. There are sentences that are not properly formulated.

 - (Lines 275 – 290). This paragraph does not include information about the results reported in this study. Therefore, it should be removed or include in the literature review section.

 - All figures included in the results section should be modified, improving their quality and resolution.

 - The structure of the Results and discussion section should be modified. Subsections should be added depending on the variable analysed.

 - (Lines 299-308). Authors should modify this paragraph. Sentences with a length of 10 lines have been included in the manuscript.

 - (Lines 377 – 291). This paragraph does not include information about the results reported in this study. The authors should move this paragraph to the methodology section.

 - (Lines 485 - 500). This paragraph does not include information about the results reported in this study. It should be moved to the methodology section.

Comments on the Quality of English Language

Moderate editing of English language is required

Author Response

Response to Reviewer Comments

Title: "Indoor Environmental Quality Assessment of Train Cabins and Passenger Waiting Areas. A case study of Nigeria"

Dear Reviewer,

Title: "Indoor Environmental Quality Assessment of Train Cabins and Passenger Waiting Areas. A case study of Nigeria"

We thank you, once again, for your insightful comments and suggestions regarding our manuscript. We appreciate the time and effort you have put into reviewing our work. Below, we address each of your comments in detail:

  • Revision of Figures and their nomenclatures: The main text of the manuscript has now been revised as per the advised nomenclatures for Figures 3, 5, 7, and 8, Also, these new nomenclatures, have been regularized and adopted in the main text's discussions accordingly.

We hope these revisions adequately address your concerns and improve the quality of our manuscript. We are grateful for your valuable feedback and look forward to further suggestions.

Sincerely,

John Omomoluwa Ogundiran

Reviewer 2 Report

Comments and Suggestions for Authors

This is an interesting case study for the Environmental Quality Assessment of Train Cabins and Passenger Waiting Areas in Nigeria. For this purpose, the Indoor Air Quality (CO2, PM, VOCs, NO2), Thermal, Acoustic, and Visual environments have been assessed in train passenger cabins and waiting areas, during 15 trips in the dry and rainy season of Nigeria. The results indicate deviations from IEQ requirements in many areas and hence suggested broader studies on the IEQ transport cabins of these regions. It is a good paper and only requires a minor revision to be ready for publication.

Only a minor revision is necessary:

-          Due to the use of several Roman and Greek symbols, a nomenclature must be included at the end of the paper.

Comments on the Quality of English Language

 Minor editing of the English language required

Author Response

Response to Reviewer Comments

Title: "Indoor Environmental Quality Assessment of Train Cabins and Passenger Waiting Areas. A case study of Nigeria"

Dear Reviewer,

We would like to express our sincere gratitude for your insightful feedback and constructive suggestions regarding our manuscript. We have carefully addressed each of the points raised, as detailed below:

  • Inclusion of Nomenclature: Response: We have created a nomenclature for all abbreviations and symbols. A comprehensive list is now available in the appendix accordingly.
  • Grammer and manuscript text revisions: Response: We have ensured that the main text grammar and readability are improved considerably without diminishing the scientific communications accordingly.

We trust that these revisions adequately address the concerns raised and enhance the overall quality of our manuscript. We appreciate the opportunity to improve our work and look forward to your further guidance.

Sincerely,

John Omomoluwa Ogundiran

Reviewer 3 Report

Comments and Suggestions for Authors

1. The abstract only mentions "gaps in thermal comfort, and exceedance of particulate matter (PM), noise, and illuminance levels". However, it did not elaborate on the specific gap.

2. The second paragraph of the Introduction explains the concept of IEQ but lacks relevant references.

3. The Introduction part lacks an overall summary of the research content of this paper, so it should be appropriately supplemented to make the research idea of this paper clearer.

4. The format of the headings in the first and later parts is different and should be adjusted to a uniform format.

5. The content of a recent study is explained in the Literature Review, but there is no corresponding reference.

6. The beginning of each paragraph should be standardized and uniform whether the character is indented.

7. When describing the results in Table 3, the interpretation of the meanings of different values of PMV and PPD can be appropriately supplemented.

8. Specific data of the above experimental results are not mentioned in the conclusion, which should be explained appropriately.

Author Response

Response to Reviewer Comments

Title: "Indoor Environmental Quality Assessment of Train Cabins and Passenger Waiting Areas. A case study of Nigeria"

Dear Reviewer,

We would like to express our sincere gratitude for your insightful feedback and constructive suggestions regarding our manuscript. We have carefully addressed each of the points raised, as detailed below:

  • Abstract's Specificity: Response: We have revised the abstract to provide specific details about the identified gaps in thermal comfort and the exceedance levels of PM, noise, and illuminance. This makes the abstract more informative and reflective of the study's findings.
  • Introduction and IEQ Concept: Response: We have added relevant references in the introduction to support the explanation of the IEQ concept, thereby strengthening the theoretical foundation of our study.
  • Summary in Introduction: Response: A concise summary of the research content has been added to the introduction, providing a clearer overview of the paper's focus and scope.
  • Uniformity of Headings: Response: We have standardized the format of the headings throughout the manuscript to ensure consistency and improve readability.
  • Literature Review Reference: Response: The missing references for the recent study mentioned in the literature review have been added, ensuring that all sources are properly cited. Furthermore, the Main texts were revised and restructured accordingly.
  • Standardizing Paragraph Beginnings: Response: The beginnings of all paragraphs have been standardized, with consistent indentation applied to enhance the manuscript's format.
  • Interpretation in Table 3: Response: Additional interpretation has been provided for the values of PMV and PPD in Table 3, offering a clearer understanding of their implications in the study.
  • Conclusion Specificity: Response: We have revised the conclusion to include specific data from the experimental results, thus providing a more comprehensive and conclusive summary of our findings.

We trust that these revisions adequately address the concerns raised and enhance the overall quality of our manuscript. We appreciate the opportunity to improve our work and look forward to your further guidance.

Sincerely,

John Omomoluwa Ogundiran

Round 2

Reviewer 1 Report

Comments and Suggestions for Authors

The authors have successfully addressed most of my previous comments. However, some minor comments need to be considered:

- (Line 350, 418 and 541) In Figure 3, 5 and 8. The sub-figures should be labeled as a) and b), and described in the figure caption

- (Line 522) In Figure 7. The sub-figures should be labeled as a), b), c) and d). and described in the figure caption.

Author Response

(The authors gave the same response as above.)
